# Underwater Fish Segmentation Algorithm Based on Improved PSPNet Network

**DOI:** 10.3390/s23198072

**Published:** 2023-09-25

**Authors:** Yanling Han, Bowen Zheng, Xianghong Kong, Junjie Huang, Xiaotong Wang, Tianhong Ding, Jiaqi Chen

**Affiliations:** 1College of Information Technology, Shanghai Ocean University, Shanghai 201306, China; ylhan@shou.edu.cn (Y.H.); zbw17377349137@163.com (B.Z.); hjunjie1212@163.com (J.H.); wxt_7182@163.com (X.W.); 17335885009@163.com (T.D.); 2College of Marine Science, Shanghai Ocean University, Shanghai 201306, China; jiaqichen9708@163.com

**Keywords:** fish, underwater fish segmentation, PSPNet, different scale features, fish characteristics

## Abstract

With the sustainable development of intelligent fisheries, accurate underwater fish segmentation is a key step toward intelligently obtaining fish morphology data. However, the blurred, distorted and low-contrast features of fish images in underwater scenes affect the improvement in fish segmentation accuracy. To solve these problems, this paper proposes a method of underwater fish segmentation based on an improved PSPNet network (IST-PSPNet). First, in the feature extraction stage, to fully perceive features and context information of different scales, we propose an iterative attention feature fusion mechanism, which realizes the depth mining of fish features of different scales and the full perception of context information. Then, a SoftPool pooling method based on fast index weighted activation is used to reduce the numbers of parameters and computations while retaining more feature information, which improves segmentation accuracy and efficiency. Finally, a triad attention mechanism module, triplet attention (TA), is added to the different scale features in the golden tower pool module so that the space attention can focus more on the specific position of the fish body features in the channel through cross-dimensional interaction to suppress the fuzzy distortion caused by background interference in underwater scenes. Additionally, the parameter-sharing strategy is used in this process to make different scale features share the same learning weight parameters and further reduce the numbers of parameters and calculations. The experimental results show that the method presented in this paper yielded better results for the DeepFish underwater fish image dataset than other methods, with 91.56% for the Miou, 46.68 M for Params and 40.27 G for GFLOPS. In the underwater fish segmentation task, the method improved the segmentation accuracy of fish with similar colors and water quality backgrounds, improved fuzziness and small size and made the edge location of fish clearer.

## 1. Introduction

As human beings focus increasingly on the diversity of fishery resources and marine ecosystems, an increasing number of research fields need to be combined with research on underwater fish segmentation, which is necessary for marine biology research, marine ecological protection and fishery resource management. In the research of these related fields, it is necessary to accurately obtain the shape, size and quantity of fish to provide data for further research. However, due to immature technology in the early days, the acquisition of morphological data of underwater fish is mainly based on traditional measurements after landing. The disadvantages of using traditional measurement methods are obvious; for example, manual measurement is time-consuming and labor-intensive and is even less efficient when a fish population is large. With the popularization and application of information technology, morphological measurements of fish have begun to be performed by transmitting machines, and there is still room for further improvement in efficiency. Therefore, underwater fish segmentation is an important research topic in the era of intelligence.

However, for special underwater scenes, the existing segmentation methods face many challenges. First, imaging equipment may encounter color distortion, noise pollution and insufficient light propagation in an underwater environment, which results in low recognition and contrast in the obtained underwater images. Second, the variety of organisms that exist in the underwater environment similar in shape and color to fish also interferes with our ability to divide fish. In addition, there is a wide variety of fish species underwater, and images of fish captured using imaging equipment tend to vary in scale and attitude. These problems make it more difficult to achieve accurate underwater fish segmentation.

At present, fish segmentation methods are divided into traditional methods and deep-learning-based methods.

The traditional method divides the target according to the edge and color of the creature’s body. In 2000, Angelo Loy et al. [1] used Fourier analysis to detect the shapes of finfish, which is considered the best automated method for detecting fish via traditional methods. In 2011, Meng-Che Chuang et al. [2] used histogram backprojection on dual local-threshold images to ensure further effective fish segmentation. In 2014, LAN Yongtian et al. [3] obtained dichotomous images of fish movements by combining three frames of difference with logical and mathematical morphological operations. In 2020, HE Qianhan et al. [4] developed a method to extract the contours of horny jaws using the Canny algorithm, contributing to easy access to information on biomorphology. In 2021, Hitoshi Habe et al. [5] proposed a National Aeronautical Advisory Council (NACA) attitude estimation method for fish wing models to identify fish accurately. However, this method required extracting fish morphology from the dataset. Moreover, if an image was blurred or disturbed by other factors, such as illumination, the extracted features were easily incomplete.

Semantic segmentation technology based on deep neural networks is a very advanced underwater fish segmentation method. Semantic segmentation belongs to a category of image classification, but it marks different image areas according to the semantic categories in the image and classifies each pixel in an image to generate the fine-grained mapping of semantic labels to image information. In 2017, Alfonso B. Labao et al. [6] used the ResNet-FCN network model to semantically segment fish in underwater videos only using input characteristics based on fish color. In 2020, Rafael Garcia et al. [7] used the Mask R-CNN architecture to locate and segment fish in images. In 2020, Fangfang Liu et al. [8] introduced an unsupervised color correction module (UCM) based on the DeepLabv3+ network and altered the upper sampling layer in the network, showing that their method improved segmentation accuracy. In 2021, Wenbo Zhang et al. [9] proposed a two-pool polymerization attentional network to improve underwater fish segmentation accuracy using a pool polymerization positional attention module and a pool polymerization channel attention module. In 2022, Jinkang Wang et al. [10] proposed an underwater image semantic segmentation method to precisely segment targets; however, the first step in this method was to improve image quality by performing image enhancement operations based on multispatial transformation. In recent years, increasing numbers of researchers have begun to improve segmentation accuracy from the perspective of integrating multiscale features of fish targets, such as the multiscale CNN network [11,12,13,14] and the porous GAN network [15,16,17,18].

Each of the above deep-learning-based approaches has its advantages, but the fuzzy and distorted images of fish in underwater scenes and the disturbance in the surrounding environment still pose challenges to accurate fish segmentation. In response to the above challenges, this paper uses PSPNet [19], which can integrate multiscale features as the basic network. An improved underwater fish segmentation algorithm based on PSPNet is proposed. This method can improve fish segmentation accuracy for fuzziness, similar colors and backgrounds and small sizes. The main contributions of this article are as follows:(1)We propose an improved PSPNet network model for underwater fish segmentation. In the feature extraction phase, this article uses an iAFF module to connect to ResBlock. iAFF realizes the full perception of multiscale characteristics and environmental information of a target through the MS-CAM module, and global and local features are integrated through AFF. In addition, iAFF integrates more contextual information through its iterative nature to facilitate the overall understanding of fish in underwater images, thereby improving fish segmentation accuracy.(2)To retain more fish characteristic information at the feature extraction stage, this article replaces the average pooling in the backbone ResNet50 network with SoftPool. Softpool reduces the number of parameters and increases the number of calculations after adding iAFF to the model through the fast exponential weighted calculation method, and the inference speed and precision are effectively improved.(3)To make fish features more distinctive in underwater environments, in this paper a triplet attention mechanism module, triplet attention (TA), is added to the different scale features in the pyramid pool module to realize more detailed attention to fish features. The TA module captures richer feature information of fish targets in a cross-latitude, interactive way, which improves segmentation accuracy.(4)In adding TA modules, we use a parameter-sharing strategy, which can reduce the numbers of model parameters and calculations by sharing the parameter weights of different scale features after passing through the TA module.

Compared to other underwater fish segmentation methods, the proposed IST-PSPNet (iAFF + SoftPool + TA) method achieves better segmentation accuracy for DeepFish datasets. In addition, the Params and GFLOPS model results do not increase compared to the baseline (PSPNet). The results also show that the proposed method can improve the MPA and FPS. In addition, we verify the effectiveness of our method.

The rest of this paper is arranged as follows. In the second section, the structure of the underwater fish segmentation IST-PSPNet method is introduced in detail. The third section gives the experimental results and analysis. The fourth section summarizes the work in this paper.

## 2. Proposed Method

### 2.1. Overall Network Structure

In this paper, a new method, IST-PSPNet, is proposed to solve the underwater fish segmentation problem. Its structure consists of an input image, a backbone network, an improved pyramid pooling module and an output image. Figure 1 shows the overall structure of the IST-PSPNet network. In this model, the main improvements were based on the ResNet50 backbone; the iAFF module was designed and connected to ResBlock in ResNet50 to achieve iterative attention feature fusion; SoftPool was used to reduce the numbers of parameters and computations, replacing AvgPool after the last ResBlock in ResNet50; and the TA triad attention mechanism module was added to the different scale features in the gold-tower pool module to focus on the specific location of fish body features in the channel.

In the IST-PSPNet network structure for the input underwater fish images, feature extraction is carried out by the improved ResNet50. To mitigate the impact of the scale change in the feature map and smaller fish bodies in the feature extraction stage, connecting and embedding the iAFF module can effectively integrate fish body features with inconsistent semantics and scales by aggregating contextual information from different receptive fields for fish bodies with different scales. After that, the feature map can retain more feature information while reducing the size through SoftPool.

Then, the resulting global feature information is passed through the pyramid module, and four different sizes of feature information are obtained using different degrees of pooling operations. After that, the feature information of four different sizes is passed into the TA module through convolution. While the parameter number is ignored, the TA module captures the cross-dimensions between the spatial and channel dimensions of the first first-scale feature via cross-latitude interaction. Then, the first first-scale feature is shared with other features via the weight of the TA module through a parameter-sharing strategy, which reduces the number of parameters while improving the overall generalization ability.

Finally, the feature information obtained through the TA module is upsampled and then interpolated with the original feature map. The feature information extracted from the shallow network is fused with the deep feature information after passing through the pyramid network to obtain the global feature information. Finally, the fish prediction results are obtained after decoding.

### 2.2. iAFF Module

During the fish feature extraction stage, the ResBlock in ResNet50 utilizes skip connections, a form of linear connection, to pass the input feature information directly to the ResBlock so that the input feature information is directly added to the output. However, this approach does not fully perceive the context and does not further improve the semantic and scale inconsistencies between input features. For special scenes, such as those underwater, the feature information needs to be more detailed. Therefore, we proposed that the iterative attention feature fusion (iAFF) module [20] (shown in Figure 2) replace the common fusion approach in ResBlock. Experiments show that embedding this module can improve fish segmentation accuracy in underwater scenes.

To fully perceive the context, initially integrating the input features is the key point. In this paper, AFF is used to implement two attention modules to fuse input features to form an iAFF (iterative attention feature fusion) module. The equations for AFF and iAFF are as follows:(1)Z=M(X⊎Y)⊗X+(1−M(X⊎Y))⊗Y
(2)X⊎Y=M(X+Y)⊗X+(1−M(X+Y))⊗Y

*X* is the constant mapping of input characteristics, *Y* is the residual of learning in ResNet, *Z* is the output fusion feature, ⊗ denotes elementwise multiplication and ⊎ is the integration of initial input characteristics. The above calculation equations represent the process of combining the different initial features *X* and *Y*. 1 − *M* (*X* ⊎ *Y*) is the dotted line in the iAFF structure; the fusion weight *M* (*X* ⊎ *Y*) is made up of real numbers between 0 and 1; and 1 − *M* (*X* ⊎ *Y*) is made up of real numbers between 0 and 1, which enables the model to learn the weight between *X* and *Y* via training. *M* denotes the multiscale channel attention module, which is the core module that makes up AFF and iAFF and whose structure is shown in Figure 3. The key idea is to achieve channel attention at multiple scales by changing the size of the spatial pool.

MS-CAM aggregates more contextual information along channel dimensions, adds global mean pooling as a global channel branch and selects point-by-point convolution (PWConv) as a context aggregator for local channels. Compared to other channel attention modules, MS-CAM can simultaneously focus on larger objects with a more global distribution and smaller objects with a more local distribution. For this particular underwater scenario, the enhanced model’s attention to the local features of smaller fish is undoubtedly critical.

### 2.3. SoftPool

During the feature extraction phase, the ResNet50 used in this article reduces the size of the feature diagram via pooling, a process that is important for increasing the sensory field and reducing the computational load. However, in underwater scenes, images are often affected by complex factors such as light, sediment and water quality, which lead to some ambiguity, distortion and fish feature distortion. Therefore, the pooling operation, which can retain more characteristic information, is the key to fish edge feature extraction.

In this paper, we preserve more characteristic information in the feature diagram during the poolization process. We replace AvgPool in ResNet50, the backbone of PSPNet, with SoftPool [21], a rapid exponentially weighted activation sampling method derived from ablation experiments using a range of architectures and pool-based methods, as shown in Figure 4. In the ImageNet 1K classification task, replacing the pool layer in ResNet50 revealed that SoftPool showed some improvement in accuracy and CUDA-based inference time (FPS) and reduced computational complexity (GFLOPS) compared to architectural baselines and other pool methods.

Softpool is a kernel-based pooling approach that provides a balance between maximum and average pool-based operations by exponentially weighting each part of a region based on the strength of the feature map region. In our experiments, we demonstrated that SoftPool can retain more information about fish features on the feature maps of underwater scenes, which is directly reflected in improved segmentation accuracy and improved computational and memory efficiency.

As shown in Figure 4, the SoftPool working diagram is subsampled using a (2 × 2) kernel to output an exponential weighted sum of the original pixels in the fish feature area.

This greatly improves the representation of the high-contrast areas that exist at the edge of the fish and in underwater scenes. To simplify the symbols, we ignore the channel dimension and assume that R is the index set corresponding to the features in the considered 2D spatial region. The *Wi* weight is the ratio of the natural index of a feature to the sum of all the features. Here is how SoftPool calculated this:(3)$$\tilde{a}=\sum_{i\in R}\frac{e^{a_{i}}\times a_{i}}{\sum_{j\in R}e^{a_{j}}}$$

$\tilde{a}$ is the SoftPool output value. $a_{i}$ denotes each feature. $\sum_{j\in R}e^{a_{j}}$ is the sum of the natural indices of all the features.

### 2.4. TA Module

In underwater environments, due to the different distances traveled by light of different wavelengths in water, water colors vary in images. As a result, there are situations where fish of different colors and water with different coloration exhibit highly similar features. As shown in Figure 5a, the textured colors of the three tagged fish are highly similar to the background colors, which makes it difficult to identify fish body features. In addition, underwater environments typically contain ecological information such as algae, vegetation and reefs, which also directly contribute to the low differentiation of fish in such scenarios. As shown in Figure 5b, there are reefs and vegetation interference in the living environment of the three species of fish, which limit the attention of the lower network to the characteristic fish information. In the fish segmentation task, we believe that adding attention mechanisms after different scale features can improve the adaptability and robustness of a network to the underwater environment, placing focus more on the detailed fish feature information and improving the understanding ability of fish features.

In the feature extraction stage, the backbone network (ResNet50) utilizes the iAFF module’s MS-CAM channel attention module to continuously acquire global and local feature weights along the channel dimension of the feature maps. This enhances the information exchange among channels. However, weighting in spatial dimensions is neglected, which means that the model cannot accurately adjust spatial feature responses in different locations as the layers deepen. Additionally, when calculating channel dimension weights, the global mean pooling performed breaks down the space in the input feature diagram into one pixel per channel. This results in a large loss in spatial information so that, when calculating attention on these single-pixel channels, there is no interdependence between channel and spatial dimensions. This may affect segmentation performance in fish segmentation missions.

To solve the problem of fish features being obscured by water quality and other ecological information interference, a triplet attention mechanism (TA) module [22] is added to the pyramidal pool module (PPM) after different scale features. The TA module constructs the dependence of fish features between channels and spatial dimensions by rotating operation and residual transformation. The channel and spatial information are encoded with a negligible number of parameters. Specifically, for fish features, the TA module can focus spatial attention on specific locations in the channel by interacting across dimensions. The use of a TA module can compensate for our lack of attention to spatial dimensions by suppressing background interference in underwater scenes, highlighting fish features (contours and details) and making the different scale features in the pyramidal pool module more distinguishable, which can then be fused for more detailed results.

As Figure 6 shows, the input of the triad attention mechanism module is a small-scale feature in a pyramid pool module consisting of three parallel branches, in which *Z*-pool is responsible for reducing the tensor’s zero dimension to two dimensions by connecting the average pool feature and the maximum pool feature on the tensor dimension in that branch. This enables the layer to retain a wealth of information in the original tensor while reducing its depth for further calculation, as represented by the following equation for *Z*-pool:(4)$$Z-Pool(x)=[MaxPool_{0d}(x),AvgPool_{0d}(x)]$$
Of these, 0d is the zero dimension after maximization and average poolization. For example, a *Z*-pool with a tensor in the shape of (C × H × W) produces a tensor in the shape of (2 × H × W).

In this module, the interaction between the height dimension and channel dimension is first established at the top branch; the input tensor X rotates counterclockwise at 90° along the height axis H to obtain the tensor ${\overset{\wedge }{X}}_{1}$; then tensor ${\overset{\wedge }{X}}_{1}$ is reduced to ${\overset{\wedge }{X}}_{1}^{*}$ via the *Z*-pool back channel and standard convolution of 7 × 7 through the core. Then, the tensor is passed to the sigmoid activation layer via the normalized layer to generate the final effective attention weight. The attention weight is then applied to ${\overset{\wedge }{X}}_{1}$ and rotated clockwise at 90° along the height axis H to obtain the original input tensor shape. Similarly, input on the second branch rotates counterclockwise at 90° along the width axis W to obtain tensor ${\overset{\wedge }{X}}_{2}$, reduces to ${\overset{\wedge }{X}}_{2}^{*}$ after passing through *Z*-pool and then undergoes a series of operations identical to the first branch to obtain the final attention weight. In the last branch, the input tensor X passes through the *Z*-pool to obtain ${\overset{\wedge }{X}}_{3}$ and experiences the normalization layer and the sigmoid layer to generate the final attention weight. The final output is then obtained via averaging operations to aggregate the fine tensor of the shape (C × W × H) generated by each of the three branches. Therefore, the above process can be expressed in the following equation:(5)$$y=\frac{1}{3}\bar{\left({\overset{\wedge }{X}}_{1}\sigma (\delta_{1}({\overset{\wedge }{X}}_{1}^{*})\right)}+\bar{\left({\overset{\wedge }{X}}_{2}\sigma (\delta_{2}({\overset{\wedge }{X}}_{2}^{*})\right)}+X\sigma (\delta_{3}({\overset{\wedge }{X}}_{3}))))$$
σ represents the sigmoid activation function. $\delta_{1}$, $\delta_{2}$ and $\delta_{3}$ represent a two-dimensional convolution layer with a kernel size of (7 × 7) in each of the three branches.

In this paper, after adding the TA module to the different scale features of the pyramid pool module, we also use a parameter-sharing strategy to share the weight of features learned from small-scale features through the TA module to other scale features to improve the generalizability and robustness of the input of the model. In summary, the addition of the TA (triplet attention) module aims to enhance the focus on fish characteristics within the pyramid pooling module without increasing the number of parameters, thereby improving the accuracy of the fish segmentation model. The parameter-sharing strategy is employed to accelerate the model-training process and inference speed, thereby enhancing the computational efficiency of the model.

## 3. Experiments

### 3.1. Experimental Setting

#### 3.1.1. Dataset

Our method was experimentally validated using the DeepFish dataset. The DeepFish dataset, collected by James Cook University’s Alzayat Saleh team from 20 habitats in the remote coastal marine environment of Australia’s subtropics, contains approximately 40,000 images captured using fixed underwater cameras. The team’s main goal was to study the effects of underwater habitat characteristics and environmental background on fish biota. The dataset consists of three parts: classifying, localization and segmenting. Classifying is for image classification tasks, while localization is for target detection tasks. Our task used the segmentation section, which contained fish segmentation labels. This section deals only with foreground and background types of fish in the water. Segmentation consists of two types of images of habitat without fish and habitat with fish, with a total of 620 images and their corresponding segmented masks. For our experiment, the quality of model training was affected by the fact that 310 images did not contain fish targets. Therefore, we extracted images of 20 habitats using Python code, with 25 images containing fish extracted for each habitat. We specifically created corresponding labels to match the images and label masks. In this work, we combined the extracted images with the previous 620 images to obtain a total of 1120 images, of which the training set contained 784 images, and the verification set and the test set had 168 images each. In this dataset, there were various pieces of ecological information and water-color-specific situations. In this case, the fish had the typical characteristics of fuzzy images, unclear feature textures, low contrast, occlusion, etc., which created certain difficulties in fish segmentation. To verify the robustness of the experiment, the dataset was not subjected to any image enhancement operation to bring it closer to a pristine environment.

#### 3.1.2. Implementation Configuration

This article trained our proposed model with NVIDIA GeForce RTX 3060, which is based on Windows 10, Python 3.8 and PyTorch 1.2. Specific experimental configurations are shown in Table 1.

In this article, the dataset was divided into training sets, validation sets and test sets at 70:15:15. During the data-preprocessing phase, we unified the size of the images through greyscale filling to ensure a true proportion of fish features, and we selected a size of 480 × 480 as the uniform image input size.

**During training**. This article’s epoch was set to 100, the batch size was set to 4 and the learning rate was set to $5\times 10^{-4}$. In addition, we chose “Adam” as the optimizer, with the Momentum parameter within “Adam” set to 0.9 and the learning rate attenuation mode set to “Cos.”

**Evaluation indicators**. Miou, Params and GFLOPS were selected to evaluate model performance, and accuracy was selected to evaluate other methods. Miou refers to the mean IoU, which represents the intersection between the segmentation results of each category and the real mask. Params refers to the number of trainable parameters in a network model, which represents the spatial complexity of a network model as a whole expressed in millions (M). GFLOPS represents a network model with a billion floating point operations per second, representing the time complexity of the network model as a whole in gigabytes (G). Accuracy is the ratio of pixels correctly predicted by a network model to the total pixels in a given category.

### 3.2. Ablation Experiments and Analysis

#### 3.2.1. Quantitative Evaluations

To validate the effectiveness of the IST-PSPNet approach presented here, we designed three ablation experiments to validate the addition of the iAFF module, the replacement with SoftPool and the addition the TA module. In this paper, we first used PSPNet with ResNet50 as the backbone network as the baseline model. Second, the iAFF module was connected to the ResBlock (named I-PSPNet) in ResNet50 via short hops to achieve multiscale feature fusion and improve underwater fish feature extraction accuracy. Third, we used SoftPool to replace the average pool operation in the ResNet50 backbone network (named IS-PSPNet) to reduce the computational volume while retaining more feature information, thus improving the segmentation performance of the network model. Finally, we added the TA module to the different scale features of the pyramid pool module and implemented parameter weight sharing (named IST-PSPNet) through the parameter-sharing strategy.

Table 2 shows the results of the PSPNet-based ablation experiments. The results showed that the PSPNet baseline model had 87.46% for the Miou, 46.70 M in Params and 45.93 G in GFLOPS.

The Miou increased by 2.08%, Params by 6.4 M and GFLOPS by 4.63 G compared to the baseline (PSPNet). The results show that the iAFF method could improve the network performance compared to the PSPNet baseline and that the MS-CAM module could combine global and local feature contextual information in channel dimensions to effectively integrate fish features at different scales. Therefore, the iAFF module could improve underwater fish feature extraction accuracy to improve fish segmentation performance. Additionally, the numbers of network parameters and calculations increased only slightly compared to the baseline. As shown in Figure 7, the effect of the addition of the iAFF module on fish segmentation is visualized in the fourth column. In the visualization of the third column baseline, some fuzzy edge features similar to the background were ignored by the baseline, but better segmentation was achieved through the addition of the iAFF module, which enabled fish body detail features to be efficiently extracted. For example, as seen from the fourth graph in the fourth column, for murky, fuzzy underwater environments, the improved I-PSPNet model adapted context-aware fusion to receive features, effectively enhancing the ability to segment fins in greater detail.

Compared to I-PSPNet, IS-PSPNet increased the Miou by 0.82%, while Params decreased by 6.62 M and GFLOPS decreased by 10.30 G. The results show that, by replacing the average pool in ResNet50 with SoftPool, more information could be retained in the feature diagram, thus improving the accuracy of fish segmentation tasks. Additionally, because SoftPool is exponentially weighted, it reduced the numbers of parameters and calculations in the model compared to average pooling. As seen in the fifth column of Figure 7, the ability to retain more information in the feature diagram was useful for capturing the edge of the body in more detail; for example, the tail and fish pectoral fins could be segmented effectively.

Compared to IS-PSPNet, IST-PSPNet increased by 1.2% for the Miou, was unchanged on Params and increased by 0.01 G for GFLOPS. The results show that the added TA module could capture richer fish features with almost no reference to the number of parameters via cross-dimensional dependencies between channel and spatial locations. As Figure 7 shows, the IST-PSPNet in the sixth column of the figure produced significantly better segmented in the visualization results. The small fish in the first picture, the small fish in the second and third pictures and the large fish in the fourth picture could be better segmented in terms of edge positioning and body detail. This suggests that attention to channel dimensions and spatial dimensions may contribute to fine-grained fish segmentation.

Overall, compared with the baseline method (IST-PSPNet), the Miou increased by 4.1%, Params decreased by 0.22 M and GFLOPS decreased by 5.66 G. The increase in all three modules contributed to the improvement in the Miou of the model. The results show that the iAFF module was very suitable for underwater fish segmentation tasks. Softpool replacement played a role in reducing Params and GFLOPS. Finally, the use of the TA module and parameter-sharing mechanism made our model achieve a better segmentation effect, while the overall parameters and GFLOPS did not change much.

#### 3.2.2. TA Module Component Effectiveness Assessment

To verify the validity of the three branches of the TA module used in this paper, we designed a set of independent experiments for the three branches that made up the TA module.

For results compared to the baseline (IS-PSPNet), ${\overset{\wedge }{X}}_{1}$, ${\overset{\wedge }{X}}_{2}$ and ${\overset{\wedge }{X}}_{3}$ increased the Miou of the model by 0.42%, 0.29% and 0.04%, respectively, as shown in Table 3. Together, ${\overset{\wedge }{X}}_{1}$, ${\overset{\wedge }{X}}_{2}$ and the two branches acted on the model, increasing the Miou by 0.53%. Of these, ${\overset{\wedge }{X}}_{3}$ had the most significant enhancement, suggesting that spatial attention construction of fish features was the most conducive to improved segmentation performance. Finally, we combined ${\overset{\wedge }{X}}_{1}$, ${\overset{\wedge }{X}}_{2}$, ${\overset{\wedge }{X}}_{3}$ and three constitutive TA modules to achieve the best segmentation accuracy, resulting in an increase of 1.2% compared to the baseline. Experiments in this group demonstrated the validity of the three branches of the TA module and the TA module proposed in this paper.

#### 3.2.3. Evaluation of Average Pixel Accuracy and Reasoning Speed

Figure 8 shows a comparison of the average pixel accuracy (MPA) of the four ablation experimental models across 100 training cycles. The MPA calculates the proportion of pixels in the forecast to the total number of pixels by comparing the pixels in the forecast with the corresponding label mask. Furthermore, the closer the MPA value is to 1, the better the segmentation performance of the model. As can be seen in the diagram, MPA was generally progressive with each method substitution addition. Of these, the IS-PSPNet MPA results were lower than the baseline (PSPNet) between 40 and 80 epochs, possibly because the model had not learned enough semantic information about the data after adding SoftPool.

As shown in Table 4, our proposed method (IST-PSPNet) had a reasoning speed of 11.68 with RTX 3060 in our experiments. First, for the baseline (PSPNet), we added iAFF and replaced the common summation operation in ResBlock with feature fusion, which increased computational efficiency and improved the FPS by 0.33.

Second, replacing AvgPool with SoftPool resulted in an increase in FPS of 1.3 for the method (IS-PSPNet). SoftPool utilizes exponential weighting, which offers higher parallelism in feature-pooling calculations. It also applies the softmax function to normalize feature maps, reducing computational redundancy and achieving higher FPS via weight calculation for each position.

Finally, the TA module was added to the IS-PSPNet method. Because the TA module captured rich information between fish features in an interdimensional interaction between independent branches, it could provide a significant performance with a negligible number of parameters. Then, we used the parameter-sharing strategy to reduce the number of parameters and storage requirements of our model by sharing parameter weights to improve the model’s reasoning speed. Therefore, our final method (IST-PSPNet) obtained the highest inference rate of 11.68.

#### 3.2.4. Visualization of the Heatmap

To represent the visualization results of different modules added to our experiments, we performed visualization (Grad-CAM) with different target layers. As shown in Figure 9b, after adding the iAFF module, we visualized and found that, while there was a focus on fish targets, there was still no positioning on some edge features. Additionally, there was some attention to background. As shown in Figure 9c, after replacing the SoftPool pooling operation, we found that the fish edges were positioned more accurately, and there was less focus on the background. As shown in Figure 9d, the visualization result of the final TA target layer showed that the figure did not focus on other regional information and was only located on the fish body. There was also a high degree of attention to the outline edges of fish. At the same time, in the figure, the distinction between the background and foreground fish was more obvious. This further verified that the improved method proposed in this paper can effectively enhance the edge details of different fish bodies and make them accurately segmented in a fuzzy underwater environment.

### 3.3. Comparison Experiments and Analysis

#### 3.3.1. Comparison with Other Popular Methods

To further validate our approach, we compared our approach to other popular semantic segmentation methods with the DeepFish dataset. We used ResNet50 as a backbone network in our experiments, with the same set of other parameters and the same training sets, validation sets and test set ratios. Here, we used a background assessment and Miou metrics to comprehensively evaluate the model’s performance. Specific results are shown in Table 5.

Compared to other popular methods, our approach (IST-PSPNet) demonstrated significant advantages. By comparing the mean intersection over union (Miou), our method outperformed FCN, UNet, SegNet and DeepLabv3+ by 6.02%, 4.24%, 3.31% and 2.88%, respectively. This indicates that IST-PSPNet effectively utilized the multiscale feature fusion mechanism of the iAFF module, retained more information in feature pooling with SoftPool and improved segmentation performance through cross-dimensional interactions of channel and spatial attention in the TA module.

By comparing the background accuracy, we observed that even the FCN’s Miou with poor results was 98.63% on background accuracy. This is because, for our underwater fish segmentation mission, we had only two categories: one background and one foreground fish. However, with regard to background, we made up the vast majority of the images. As a result, each method had a higher segmentation result for background accuracy, but our method (IST-PSPNet) achieved 99.78% for background accuracy, revalidating the advantages of our approach.

By combining background accuracy and Miou, we can conclude that the proposed IST-PSPNet achieved an excellent segmentation performance for each category, as well as overall.

#### 3.3.2. Prediction Visualizations of Different Methods

Figure 10 shows the predicted results of different semantic segmentation methods for the DeepFish dataset, and our method (IST-PSPNet) showed a significant advantage in fish segmentation compared to other popular methods.

For the first picture in the first column, we looked at its corresponding label mask (the first picture of the second column) and could see that it had the delicate feature of a fish palpus. In the results of other methods, this fine feature was not segmented, and our method (IST-PSPNet) achieved this fine feature segmentation.

For the second, third and fourth images in the first column, FCN, UNet, SegNet and DeepLabv3+, while successfully segmenting the fish, were imprecisely located in detail, such as the edge of the fish body. These fish were similar to the background in this underwater scene and came with ambiguities. For example, in the fourth image of the first column, even humans have some difficulty identifying with the naked eye. In contrast, our proposed method (IST-PSPNet) achieved precise segmentation and edge detail for fish. For example, in the fourth image of the first column, no other method could separate the caudal fins of the fish, and our method (IST-PSPNet) could effectively segment edge details such as caudal fins.

## 4. Conclusions

Currently, acquiring underwater fish morphology data is still mostly based on traditional measurements after fishing and bringing the fish ashore. To improve efficiency, most technologies are now moving toward underwater real-time fish segmentation, which makes it easier to obtain fish morphology data. However, due to the influence of underwater water quality and the influence of many other types of ecological information, the images are blurred and distorted. To solve these problems, a high-precision segmentation method (IST-PSPNet) was proposed. Experiments showed that, compared to other semantic segmentation methods, this method could improve the segmentation accuracy for small fish with blurring and similar colors and backgrounds. Moreover, in underwater fish segmentation, our method achieved good segmentation accuracy.

(1)To fully relate the extracted features of different scales to the context in the feature extraction stage, we proposed an iterative attention feature fusion method based on an iAFF module. Through this method, we realized the depth mining of different scale feature information. Moreover, for this particular underwater scenario, this method could effectively integrate local feature information and global feature information to achieve full awareness of context information. In addition, we also solved the problem of how to initially integrate the received features through iteration.(2)In an underwater environment, extracting more information about the characteristics of fish can help to better segment them. In this paper, the average pool in the backbone ResNet50 network was replaced by SoftPool to address the lack of feature information caused by the pooling process. In addition, SoftPool calculated features in a rapid, exponentially weighted way. In this way, compared with average pooling, the numbers of parameters and calculations were reduced to a great extent and the reasoning speed was accelerated.(3)To make the network model more suitable for fuzzy underwater scenes, we added a triplet attention (TA) module after different scale features of the pyramidal pool module. The TA module captured the specific position of fish features in the channel dimension through the spatial dimension of the independent branch to realize the attention to fish features. The underwater fish segmentation performance was improved without increasing the calculation parameters.(4)In this paper, a parameter-sharing strategy was utilized when adding the TA module. This strategy enabled different scale features in the pyramid pooling module to share the same parameters. In this way, the numbers of model parameters and calculations were greatly reduced.

The method proposed in this paper may play an important role in promoting the development of intelligent fisheries and provide some help with intelligently obtaining fish data. The focus of future research should be to reduce the numbers of parameters and computations of network models in underwater fish segmentation through further research to achieve lightweight processing.

## Figures and Tables

**Figure 1 sensors-23-08072-f001:**
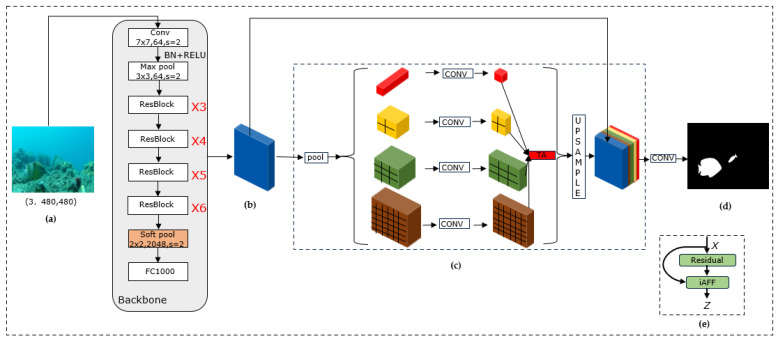
IST-PSPNet network architecture. (**a**) Input image, (**b**) extracted original feature map, (**c**) improved pyramid pooling module, (**d**) final output image prediction and (**e**) iAFF-ResBlock jump connection diagram.

**Figure 2 sensors-23-08072-f002:**
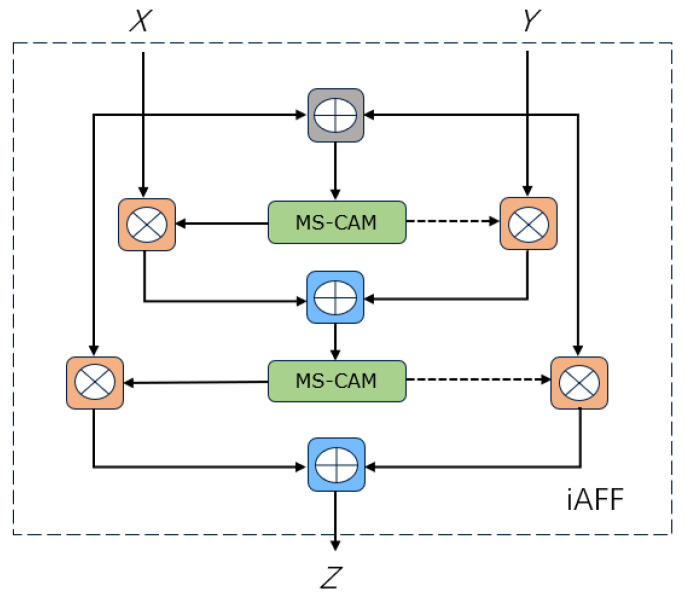
iAFF module structure.

**Figure 3 sensors-23-08072-f003:**
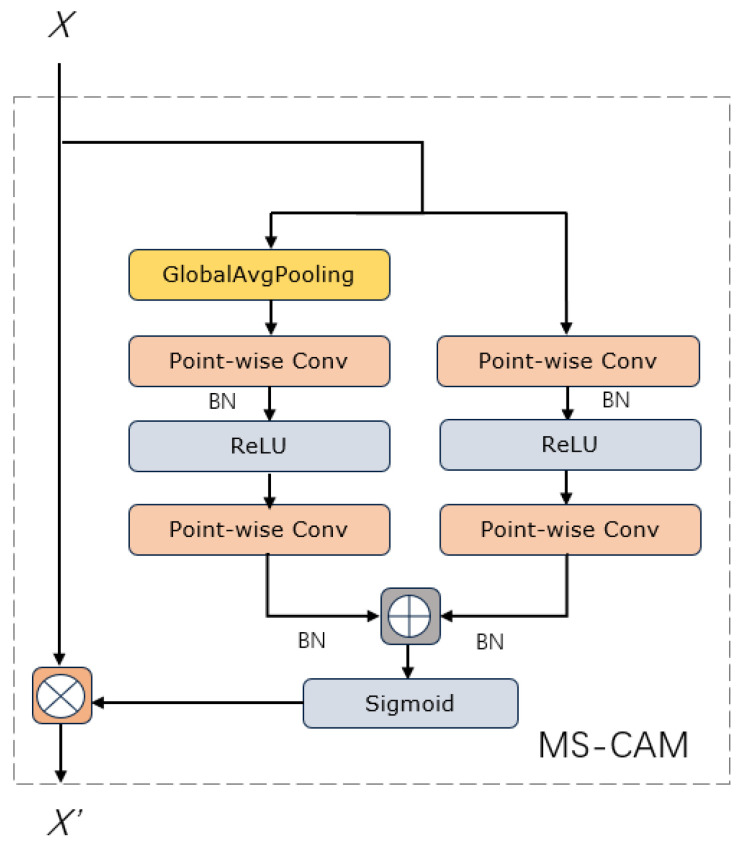
MS-CAM module structure.

**Figure 4 sensors-23-08072-f004:**
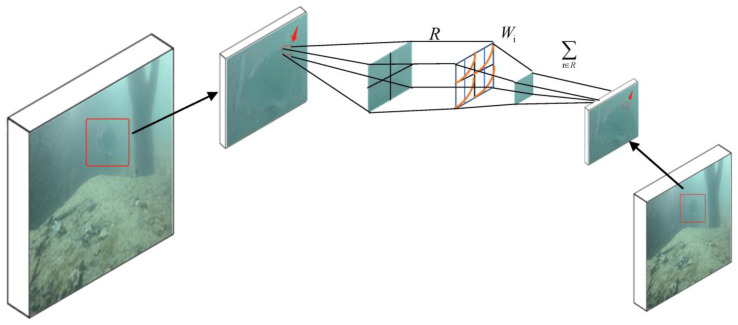
Softpool illustration.

**Figure 5 sensors-23-08072-f005:**
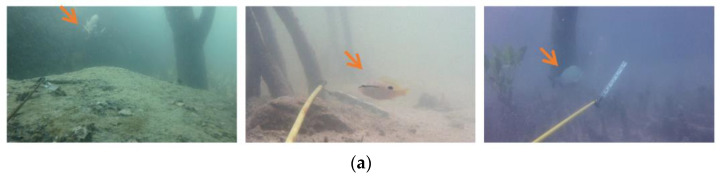
The underwater fish scenes in the DeepFish dataset. (**a**) The color of the fish body is similar to the color of the underwater background. (**b**) The color of the fish is similar to the color of the disturbance (stone, wood) in the background.

**Figure 6 sensors-23-08072-f006:**
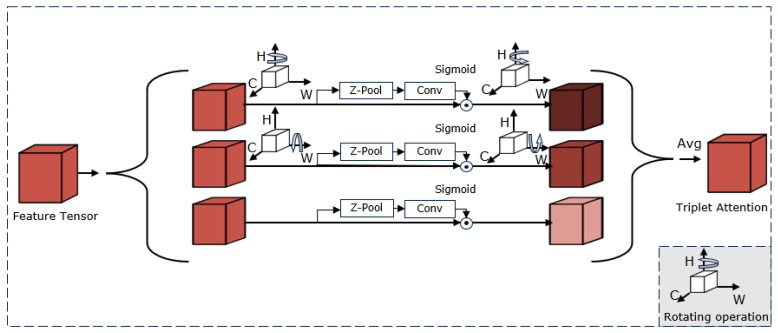
TA module structure.

**Figure 7 sensors-23-08072-f007:**
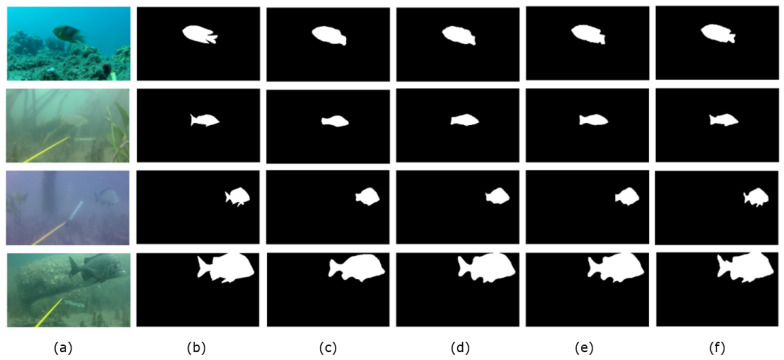
Visual prediction results of ablation experiments. (**a**) Original input image. (**b**) A mask corresponding to the original input image. (**c**) PSPNet results. (**d**) Results from I-PSPNet. (**e**) Results of IS-PSPNet. (**f**) Results of IST-PSPNet.

**Figure 8 sensors-23-08072-f008:**
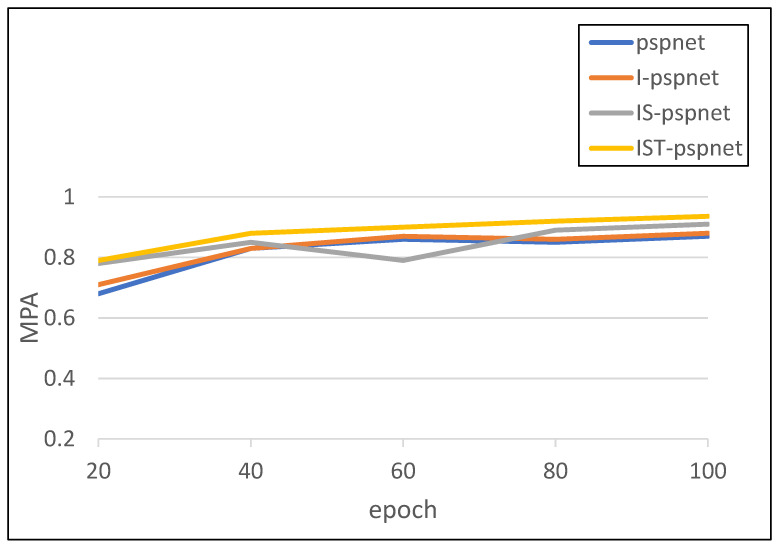
Comparison of average pixel accuracy with improved network model.

**Figure 9 sensors-23-08072-f009:**
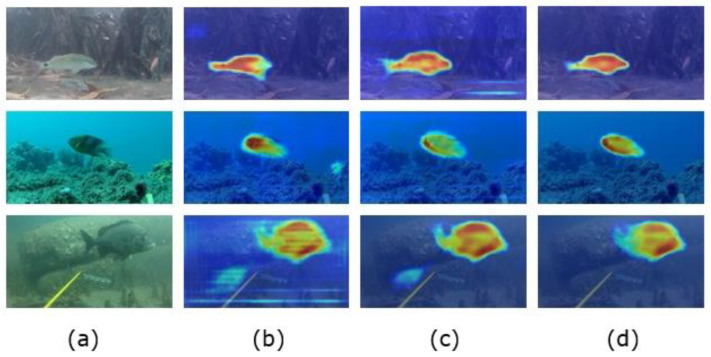
Visualization of different module locations for the DeepFish dataset. (**a**) Original image and (**b**) Grad-CAM results at the iAFF module location after adding the iAFF module. (**c**) Grad-CAM results for the last layer of the backbone network. (**d**) Grad-CAM results at the location of the TA module after adding iAFF module, SoftPool module and TA module.

**Figure 10 sensors-23-08072-f010:**
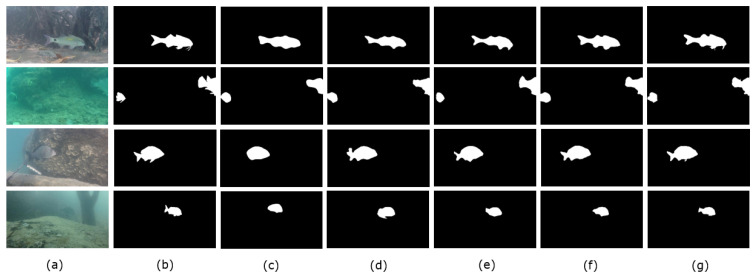
Qualitative comparison of different semantic segmentation methods for DeepFish dataset. (**a**) Raw images. (**b**) Label masks. (**c**) FCN results. (**d**) UNet results. (**e**) SegNet results. (**f**) Deeplabv3+ results. (**g**) IST-PSPNet results.

**Table 1 sensors-23-08072-t001:** Experimental configuration of this paper.

Environment	Version
CPU	Intel i7-11700, 2.50 GHz
GPU	NVIDIA GeForce RTX 3060
OS	Windows 10
CUDA	V 11.6.1
Python	V 3.10
Torch	V 1.13.1

**Table 2 sensors-23-08072-t002:** Results of ablation experiments.

Model	Backbone	iAFF	SP	TA	Miou	Params	GFLOPS
PSPNet	ResNet50				87.46%	46.70 M	45.93 G
I-PSPNet	ResNet50	√			89.54%	53.10 M	50.56 G
IS-PSPNet	ResNet50	√	√		90.36%	46.48 M	40.26 G
IST-PSPNet	ResNet50	√	√	√	91.56%	46.48 M	40.27 G

**Table 3 sensors-23-08072-t003:** Impact of Three TA Branches on Models.

TA:Branch	Miou/%
Basic (IS-PSPNet)	90.36%
${\overset{\wedge }{X}}_{1}$	90.78%
${\overset{\wedge }{X}}_{2}$	90.65%
${\overset{\wedge }{X}}_{3}$	90.96%
${\overset{\wedge }{X}}_{1}$ $+{\overset{\wedge }{X}}_{2}$	90.89%
${\overset{\wedge }{X}}_{1}$ $+{\overset{\wedge }{X}}_{2}$ $+{\overset{\wedge }{X}}_{3}$	91.56%

**Table 4 sensors-23-08072-t004:** Comparison of model reasoning speed.

Models	FPS
PSPNet	8.23
I-PSPNet	8.56
IS-PSPNet	9.86
IST-PSPNet	11.68

**Table 5 sensors-23-08072-t005:** Quantitative results for different segmentation methods for DeepFish dataset.

Model	Backbone	Background Accuracy	Miou
FCN	ResNet50	98.63%	85.54%
UNet	ResNet50	99.12%	87.32%
SegNet	ResNet50	99.14%	88.25%
Deeplabv3+	ResNet50	99.60%	88.68%
IST-PSPNet	ResNet50	99.78%	91.56%

## Data Availability

Enquiries regarding the experimental data should be made by contacting the first author.

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
