# Peer review of "Underwater Fish Segmentation Algorithm Based on Improved PSPNet Network"

_sensors, 2023, doi:10.3390/s23198072_

Round 1

Reviewer 1 Report

I report a review of the paper entitled "Underwater Fish Segmentation Algorithm Based on Improved PSPNet Network." This paper develops a method for semantic segmentation of underwater fish in videos acquired using an underwater camera. This technique is very versatile and important in a very wide range of applications. The proposed method is interesting because it is an improvement of PSPNet. However, this manuscript has a number of problems with the English writing and formatting. There are many unclear and flawed descriptions, and the manuscript in general needs to be proofread by an experienced person or service.

Line 12-17: The description below is too redundant. It is hard to believe that this is a single sentence. I think a semicolon should be a period. Also, there are too many fragmented phrases.

"First, in the feature extraction phase, in order to fully perceive the characteristics and environmental context information of different scales, we propose an iAFF based iterative attention feature fusion mechanism, which realizes the depth mining of fish body features of different scales and the full perception of context information; then the full perception of context information; Then the SoftPool method based on fast exponential weighted activation is used to reduce the number of parameters and the amount of computation while parameters and the amount of computation while retaining more feature information, the accuracy of segmentation is improved and the efficiency is improved."

Other problems are in the following sentences. It is beyond my ability to appropriately point out problems with this language and I recommend the use of a proofreading service.

Line 33-36, Line 73-76, Line 90, Line 139-142, Line 145-149, Line 160-163, Line 354-358, Line 440-444

Line 59: What does the (*) in the following sentence indicate?

"In 2000, Angelo Loy * et al. [1]" 

Figure 1: The red line in the text in the figure is unnecessary.

 Line 194: The symbol with a circle and an X appears to be a Tensor product, but it is not clearly indicated. This symbol is used in various contexts, so its definition should be clearly stated. The authors use a symbol consisting of U and +, but it is not clear what this indicates. The following statement is not a standard operation. A more detailed explanation is needed.

 "integration of initial input characteristics"

 Line 196 : I think the following equation is a typo.

 1 - M (x thinning Y)

 Line 259-266: I do not understand the meaning of "b(a)" and "b(b)". 

 line 310: I think "0d" and "od" are misspellings.

 line 331: I don't understand what is the line above the equation.

 Figure 6: I think "Tsenor" in the figure is a typo.

 Line 383-384: Learning rate is set to "s", but what is this?

Line 384: "Learning rate attenuation mode set to Cos." Cosine attenuation? If so, Adam adjusts the learning rate automatically, so normally only the initial learning rate is set and not adjusted in the schedule.

 Line 532: The spelling of Grad-CAM is not consistent.

Please look at suggestions.

Reviewer 2 Report

This study presents an underwater fish segmentation algorithm based on improved PSPNet network. This Reviewer feels that is a useful work. However, there are still several issues, which need to be improved before the manuscript can be considered for publication.

1. The description in Figure 1 is not clear. In this figure, the IAFF Resblock connection is presented as a Residual IAFF connection. In the backbone link, Resblock×3 and Resblock×4 are not drawn consistently with Resblock×5 and Resblock×6.

2. How to verify the generalization of the model, as it is only trained and validated on the Deep fish dataset.

3. The Z-pool pooling of TA module describes features as 0-dimension, how to understand it.

4. Some recent underwater image processing technologies should be included in the literature review, such as (Chang, 10.3390/electronics12132882), (Gao, 10.3390/jmse9020225), and so on.

5. The layout and typesetting problems should be avoided, such as in L244-246, and the variable i should be italicized.

6. In the list of references, the names of authors (surname and first name) need to be carefully checked and corrected, including but not limited to Refs. [3], [4], [8-12] [14], [16-18].

 Minor editing of English language required.

Reviewer 3 Report

  - Method's novelty is not adequately justified. The proposed improvements to the PSPNet network need to be articulated more precisely, and their significance in addressing the challenges of underwater fish segmentation should be explicitly demonstrated.

   -  iAFF mechanism requires a more detailed explanation, including how it addresses the specific challenges of underwater image segmentation.

   - The rationale behind the choice of the SoftPool method needs to be elaborated, along with a comparison to other potential pooling methods.

   - The proposed Triplet Attention mechanism (TA) needs clearer justification, including its potential advantages over other attention mechanisms in the context of fish segmentation.

   - The parameter sharing strategy's impact on the model's performance needs to be empirically demonstrated.

  - The experimental section lacks comprehensive validation. Comparisons against state-of-the-art methods are essential to establish the superiority of the proposed IST-PSPNet.

  - The results provided lack a detailed analysis. The paper should include discussions about the implications of the obtained metrics and what they imply about the model's performance under different conditions.

   - Visual comparisons of segmentation outputs are necessary to illustrate the strengths and weaknesses of IST-PSPNet compared to other methods.

  - It needs to clearly establish how the proposed IST-PSPNet contributes to the field of underwater fish segmentation beyond existing methods.

   - The discussion on future research directions should be expanded to include more specific areas of improvement or exploration.

Paper has many longs lines which make confusion.

Round 2

Reviewer 1 Report

All issues have been addressed.

Author Response

Okay, thank you for your previous suggestion. It really helped me publish my paper. Thank you again

Reviewer 2 Report

The authors have addressed most of my comments. I think the manuscript should be accepted after the minor revisions of spelling and formatting errors.

1) The notation of symbols should be the same, for example, in Eqs. 1 and 2, X and Y are italicized, but in the subsequent text, they are orthomorphic.

2)  Authors in the reference list, the surname should be written in full, and the first name should be abbreviated. For example: [3] Lan, Y; Ji, Z; Gao, J; Wang, W. Robot fish detection based on a combination method…, and so on.

No comments.

Reviewer 3 Report

I have no more suggestions

I have no more suggestions

Author Response

Ok, thank you for the suggestions you gave before, which are very useful to my experiment progress. Thank you again